# VecDesigner: Exploring Visual Guidance and Structural Consistency for Semantic Typography

**Liu Yu** [1]  **Xingjiao Wu** [1]  **Ziang Liu** [1]  **Jiabao Zhao** [2]  **Daoguo Dong** [3]  **Liang He** [1]

## Abstract

Semantic Typography aims to visualize the meaning of an input word through the form of a character, while preserving its legibility. Existing vector-based methods, which primarily rely on text-driven optimization like Score Distillation Sampling (SDS), often produce glyphs that lack rich semantic details. Furthermore, these approaches struggle to maintain the overall structural integrity of the glyphs and frequently suffer from visual artifacts caused by intersections, compromising both legibility and aesthetic quality. To address these challenges, we propose VecDesigner , a novel optimization-based method for vector semantic typography. Specifically, we introduce Visual-Guided Score Distillation Sampling (VGSDS), which leverages text-related reference images as visual guidance to infuse the glyphs with richer and more concrete semantic details. To preserve legibility and structural integrity, we design a vector-based Procrustes loss to constrain the overall deformation of the glyph. Concurrently, we effectively mitigate the intersection problem by imposing Positional Relationship Constraints on the control points. Comprehensive experiments demonstrate that VecDesigner outperforms existing methods in both semantic expression and structural preservation, generating high-quality, expressive, and clearly legible semantic glyphs.

## 1. Introduction

Semantic Typography aims to automatically deform the shapes of individual characters within a word to visually

[1]East China Normal University, Shanghai, China [2]Donghua University, Shanghai, China [3]Fudan University, Shanghai, China. Correspondence to: Xingjiao Wu <xjwu@pharm.ecnu.edu.cn>.

*Proceedings of the 43rd International Conference on Machine Learning*, Seoul, South Korea. PMLR 306, 2026. Copyright 2026 by the author(s).

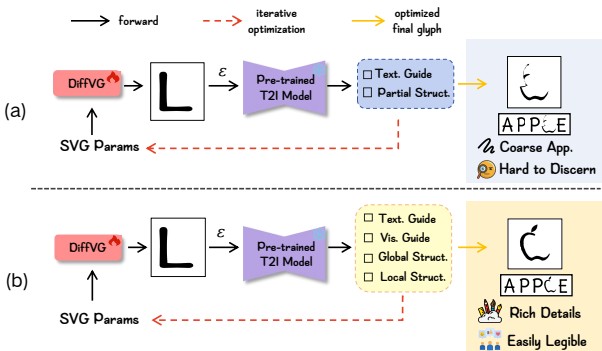

*Figure 1.* Method Comparison. (a) *Semantic Representations*: Existing method typically use vanilla SDS that relies solely on text prompts for textual guidance. As a result, they struggle to capture fine-grained details and produce glyphs with a coarse and unpolished appearance. *Glyph Structure*: The insufficient partial structural constraints adopted in prior work lead to local distortions, rendering indiscernible characters. In contrast, (b) VecDesigner incorporates additional visual guidance to generate glyphs rich in semantic details. It simultaneously preserves global structure and mitigate local artifacts to ensure high legibility.

embody its meaning while simultaneously preserving legibility. For instance, as shown in Fig.1, the letter "L" in the word "APPLE" is modified to resemble the form of an actual apple. Existing approaches to semantic typography generally fall into two categories: raster-based methods that directly generate pixelated characters (Tanveer et al., 2023; He et al., 2023; Feng et al., 2024; He et al., 2025), and vector-based methods (Iluz et al., 2023; Li et al., 2025) that synthesize characters in the Scalable Vector Graphics (SVG) format. SVG is a vector image format represented by mathematical descriptions of lines and Bézier curves. In contrast to raster graphics, vector glyphs have superior editability and resolution-independent scalability, making them suitable for artistic design. Due to these advantages, vector-based semantic typography holds significant practical value and has extensive applications across various fields, including poster design (Lin et al., 2023), education (Vungthong et al., 2017), and marketing (Xie et al., 2021), thereby deserving further academic investigation. However, effectively conveying vivid and semantically detailed visual content remains a significant challenge. Equally critical is

preserving the legibility of the deformed glyphs, an aspect that ensures structural consistency with the original ones.

From the semantic perspective, drawing inspiration from advances in text-to-image generation, recent approaches (Xing et al., 2024; 2023; Vinker et al., 2023) utilize Score Distillation Sampling (SDS) (Poole et al., 2022) to optimize their task-specific parameters by distilling semantic knowledge from the powerful pre-trained Stable Diffusion (SD) model (Rombach et al., 2022). Specifically, a vector glyph is represented by closed Bézier curves defined by control points which are optimized via the differentiable rasterizer DiffVG (Li et al., 2020). The word serves as a text prompt that provides textual guidance for generation, and SDS transfers semantic knowledge from SD into DiffVG, iteratively optimizing the glyph's parameters to align its shape with the prompt. After multiple optimization steps, we obtain the final glyph. Although vanilla SDS has demonstrated promising performance, its exclusive reliance on text prompts often limits its ability to capture fine-grained semantic details, leading to glyphs with coarse appearance that lack visual expressiveness (as shown in Fig.1(a)).

To preserve legibility, glyphs must retain their overall structural integrity. However, SDS-optimized glyphs often tend to overfit the semantics of the prompt at the expense of legibility. While efforts have been made to conform the shape of the deformed glyph with the original one, they apply local angular constraints that maintain partial structure while insufficient for holistic form. Moreover, the direct optimization of control point parameters via SDS is prone to inducing intersected Bézier curves, an issue commonly overlooked in SVG generation that causes local artifacts. Recent studies (Zhang et al., 2024; Wu et al., 2023) attempt to mitigate this problem in a data-driven way by learning latent representations from a large-scaled vector graphics datasets. While effective to some extent, such data-driven approaches are resource-intensive and misaligned with the inherently subjective and diverse nature of semantic typography as an artistic task.

To tackle the aforementioned challenges, we propose VecDesigner, a novel optimization-based method for vector-based semantic typography (as shown in Fig.1(b)). The primary goal is to synthesize vector glyphs that exhibit richer semantic details while preserving their readability and structural integrity. To enrich finer semantic nuances, we extend SDS to Visual-Guided Score Distillation Sampling (VGSDS). VGSDS leverages word-related reference images as visual guidance during the generation process. The core insight is that relying solely on text prompts often leads to abstract or ambiguous interpretations, whereas images can provide concrete visual attributes. In particular, we utilize the pre-trained Stable Diffusion model to generate reference images conditioned on the text prompt. We then formulate an energy function that, at each optimization step, minimizes the semantic difference between the features extracted from intermediate glyphs and reference images, to assist the generation process.

To better constrain the overall structural consistency of the rendered glyphs, we introduce a vector-based Procrustes loss, which can quantify the deformation distance between control points of the initial glyph and those of the generated output. Considering the sparsity of original control points, we further sample additional points along the glyph's outlines for stronger constraints. Additionally, we mitigate the intersections issue by imposing Positional Relationship Constraints on control points to restrict their spatial arrangement. We validate the effectiveness of VecDesigner through comprehensive experiments, demonstrating its advantages across various evaluation metrics. In summary, our main contributions are as follows:

- We introduce VecDesigner, a novel method for vector-based semantic typography. Compared with previous methods, VecDesigner excels in synthesizing high-quality, visually compelling and easily legible glyphs.

- We propose Visual-Guided Score Distillation Sampling to enrich semantic details of the glyph typography.

- We present a vector-based Procrustes loss to guarantee the overall deformation of glyphs and add Positional Relationship Constraints term to prevent intersections.

- Comprehensive quantitative and qualitative experiments demonstrate that VecDesigner outperforms existing methods from both semantic and structural aspects.

## 2. Related Work

### 2.1. Vector-based Semantic Typography

Our method conducts semantic typography in the vector domain. Vector representations are resolution-independent, scalable without quality loss, particularly suited for semantic typography. Early studies (Xu & Kaplan, 2007; Zou et al., 2016) investigate rearranging words or letters to form a visual image. For example, Xu et al. (Xu & Kaplan, 2007) designs a system that can partition a plain region into subregions and warp glyphs into these regions with minimal distortion. Zou et al. (Zou et al., 2016) introduce a fully automatic framework that computes the global layout for the input word and deforms the letter to fit the shape. However, these methods often give precedence to geometric shaping of a whole word at the expense of readability. More recently, Word-As-Image (Iluz et al., 2023) introduces a notable advancement by leveraging the pre-trained Stable Diffusion model for semantic typography. It employs SDS loss (Poole et al., 2022) to distill prior knowledge from SD. Dynamic

Typography (Liu et al., 2024) extends this field into the video domain by animating textual content. OracleFusion (Li et al., 2025) proposes a semantic typography framework for Oracle Bone Script decipherment. Yet, generating semantically rich, legible glyphs remains a key challenge in semantic typography. More related raster-based methods are introduced in Appendix.

## 3. Preliminary

### 3.1. Vector Fonts

Standard font formats such as TrueType (Penney, 1996) and PostScript (Adobe Systems Inc., 1990) use vector-based representations, which describe each character as outlines composed of a series of Bézier curves. Following the settings in (Iluz et al., 2023), we use the FreeType Font Library (Turner et al., 2009) to extract the outlines of a specific character from a given font file. The outlines are automatically converted into closed contours consisting of multiple Bézier curves. These curves, defined by their control points mathematically, facilitate representation of each character in Scalable Vector Graphics (SVG) format.

### 3.2. Score Distillation Sampling

Score Distillation Sampling (SDS), first proposed in DreamFusion (Poole et al., 2022), leverages a pre-trained 2D diffusion model to facilitate text-to-3D synthesis without requiring 3D data supervision. Specifically, this process begins by encoding a textual prompt into the pre-trained Imagen model (Saharia et al., 2022). Meanwhile, a Neural Radiance Field (NeRF) (Mildenhall et al., 2021) is rendered from a randomly selected camera viewpoint to generate an image $x$. SDS aims to optimize the NeRF parameters $\theta$ so that the output of NeRF aligns closely with the given textual prompt. $x$ is subsequently perturbed with Gaussian noise to obtain a noisy version $x_t = \alpha_t x + \sigma_t \epsilon$, and Imagen predicts the noise added on $x_t$. $\theta$ are optimized by minimizing the discrepancy between the actual noise and the predicted noise from the diffusion model, defined as

$$\mathcal{L}_{diff} = \mathbb{E}[w(t)\|\varepsilon_\phi(x_t = \alpha_t x + \sigma_t \varepsilon, y, t) - \varepsilon\|_2^2], \quad (1)$$

where $y$ is the text condition, $w(t)$ is a time-dependent weighting function, and $\epsilon_\phi$ is the pre-trained diffusion model. The gradient with respect to $\theta$ is given by

$$\nabla_\theta \mathcal{L}_{\text{SDS}} = \mathbb{E}_{t,\varepsilon,y} \left[ w(t)(\varepsilon_\phi(x_t, y, t) - \varepsilon)\frac{\partial x}{\partial \theta} \right]. \quad (2)$$

By iteratively applying the above procedure, NeRF is progressively refined to produce a high-fidelity 3D representation that aligns with the input text description. In this study, we adopt SDS as the foundation for semantic typography.

## 4. Method

This section introduces VecDesigner, an optimization-based method for vector semantic typography, with its framework illustrated in Fig.2. We first detail the task setting and model architecture, then describe the proposed techniques.

### 4.1. Task Formulation

In the SVG format, each character $C_{letter}$ is a vector glyph composed of Bézier curves with control points. Formally, $C_{letter}$ is defined as a path set $\{P_i\}_{i=1}^{N_p}$, where each $P_i$ consists of several drawing commands $\{C_j^i\}_{j=1}^{N_c}$, following corresponding updatable control points $\{D_k^{ij}\}_{k=1}^{N_d}$. Each control point is denoted as $D_k^{ij} = (x_k^{ij}, y_k^{ij})$, indicating its SVG canvas coordinates. In our work, we consider two command types $\{Moveto, Curveto\}$ and implement curves as cubic Bézier curves parameterized by four control points.

Our pipeline is shown in Fig.2. Given a word $W$, we aim to deform the shape of each constituent letter $C_{letter}$, enabling its final form to visually reflect the meaning of $W$ while preserving legibility. Firstly, we extract the outlines of $C_{letter}$ through FreeType Font Library and convert them to SVG format. The SVG parameters $\theta$, namely coordinates of control points, are passed to a differentiable rasterizer DiffVG $\mathcal{R}(\theta)$, to obtain a raster image $x$. At each optimization step, a VAE Encoder (Esser et al., 2021) encodes $x \in \mathbb{R}^{H \times W \times C}$ as a latent representation $z \in \mathbb{R}^{(H/f) \times (W/f) \times 4}$, where $f$ is the downsampling factor. Then, $z$ is added with Gaussian noise and turn into $z_t = \alpha_t z + \sigma_t \epsilon$, which is fed into the diffusion model. The word $W$ serves as text prompt of SD, encoded through CLIP Text Encoder (Radford et al., 2021). We propose an extension of SDS, Visual-Guided Score Distillation Sampling (VGSDS) to improve the semantic details of vector glyph by utilizing the strong prior of SD. For structural alignment, we design a vector-based Procrustes loss for global consistency and incorporate positional relationship constraint to reduce path crossovers. The gradients of the combined objective are backpropagated through DiffVG to update the control points at each iteration until convergence.

### 4.2. SVG Optimization with Visual Guidance

**Vanilla Method.** During each optimization iteration, DiffVG renders an image $x$, which is augmented and fed into the diffusion model. The gradients of SDS loss are calculated using Eq.2 and backpropagated through DiffVG to update the control points. However, vanilla SDS only conditions on a coarse word prompt, which lacks the capacity to convey nuanced semantic meanings. As a result, the synthesized images often exhibit limited fine-grained fidelity and visual quality. This motivates us to explore more information about the target word.

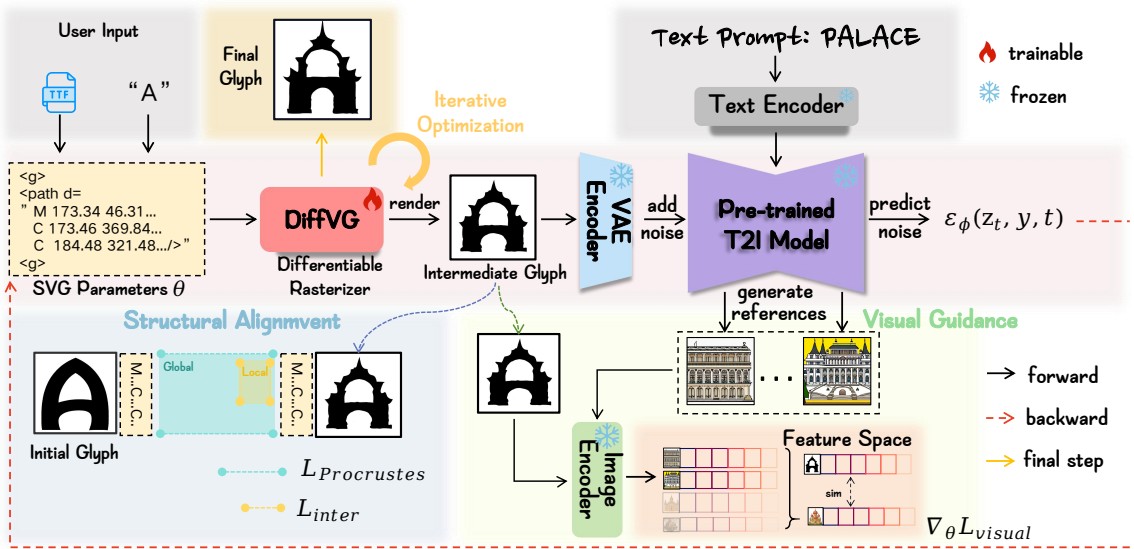

*Figure 2.* **The framework of VecDesigner.** Given a target letter and a font file, we extract its outlines and convert them to SVG format with commands and control points, which are fed into DiffVG that outputs a raster image. At each iteration, we aim to update the coordinates of control points to deform the glyph to the desired concept. The raster image is subsequently input into the pre-trained diffusion model. At the meantime, the concept is input into the model through CLIP Text Encoder. Visual-Guided SDS is employed to strengthen the semantic meaning of the concept. Structural Alignment encompasses two key components to maintain legibility: vector-based Procrustes loss that preserves the global structure, and Positional Relationship Constraints that mitigate intersections.

**How to improve semantic representation?** Inspired by recent advances in generative models (Ruiz et al., 2023; Zhang et al., 2023), images can be employed as additional conditions to strengthen guidance during generation. Unlike textual descriptions, images provide a more intuitive and information-rich appearance of semantic concepts. This multimodal conditioning framework enables finer control over the outputs, allowing the model to capture both visual and semantic subtleties aligned with the intended concept.

**How to include visual guidance?** To generate data for a specific class, Classifier Guidance (Dhariwal & Nichol, 2021) trains a classifier $p_\delta(y|z_t, t)$ and uses gradients $\nabla_{z_t} \log p_\delta(y|z_t)$ to guide the diffusion sampling process towards the desired class $y$. The predicted noise $\hat{\epsilon}$ is defined as

$$\hat{\epsilon} = \varepsilon_\phi(z_t, y, t) - s\sigma_t \nabla_{z_t} \log p_\delta(y|z_t), \quad (3)$$

where s is a hyperparameter that controls the guidance strength. From score-based viewpoint (Song et al., 2020), $\nabla_{z_t} \log p(y|z_t)$ corresponds to the Stein score, which is related to the predicted noise $\varepsilon_\phi$ by

$$\nabla_{z_t} \log p(y|z_t) = -\frac{1}{\sqrt{1 - \bar{\alpha}_t}} \varepsilon_\phi(z_t, y, t). \quad (4)$$

It highlights that the score function guides the denoising process by estimating the direction towards regions of higher data density. Motivated by this insight, we introduce an additional energy function $\mathcal{H}(z_t, t, y)$ to further steer the optimization trajectory.

Therefore, we propose to design the energy function $\mathcal{H}(\cdot)$ as a measure of semantic alignment between the intermediate result $x_{o_n}$ at optimization step $o_n$, and the visual features of the target concept, to guide the generation process. The objective of $\mathcal{H}(\cdot)$ is to encourage $x_{o_n}$ to be closer to the semantic attributes. Specifically, during the optimization process, given a textual prompt, we employ a pre-trained diffusion model to generate $N_s$ semantic images, denoted as $\{x_s\}_{s=1}^{N_s}$ at predetermined intervals. While these images may differ in visual appearance, they consistently reflect the same semantic concept. Since CLIP is pretrained on large-scale image-text pairs, its latent space is robust to noise and trivial variation. Therefore, we utilize the CLIP Image Encoder based on Vision Transformer (ViT) (Dosovitskiy et al., 2020) to extract the final feature maps $f = \{f_s\}_{s=1}^{N_s}$ from references, which share semantic representation. The semantic center is then computed by averaging these feature maps: $f_{\text{mean}} = \frac{1}{N_s}\sum_{s=1}^{N_s} f_s$. Simultaneously, the feature map $f_{o_n}$ is also extracted by the same encoder from $x_{o_n}$. The CLIP-based cosine similarity between the current feature map $f_{o_n}$ and the semantic center is formulated as

$$\mathcal{H}_{\text{clip}} = \frac{f_{\text{mean}} \cdot f_{o_n}}{\|f_{\text{mean}}\| \|f_{o_n}\|}. \quad (5)$$

In conclusion, Equation 2 can be reformulated as

$$\nabla_\theta \mathcal{L}_{\text{visual}} = \mathbb{E}_{t,\varepsilon,y}\left[w(t)\Big(\varepsilon_\phi(z_t, y, t) - \lambda_v \frac{\partial \mathcal{H}_{\text{clip}}}{\partial x} - \varepsilon\Big)\frac{\partial x}{\partial \theta}\right],$$

where $\lambda_v$ is the coefficient of $\mathcal{H}$.

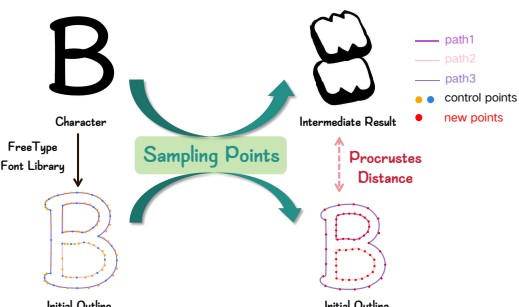

*Figure 3.* An example of vector-based Procrustes loss. Each path of the extracted outline is colored distinctly. Orange/blue dots: initial control points. Red dots: newly sampled points.

### 4.3. Structural Alignment

Besides semantic alignment, ensuring deformed glyph legibility via structure preservation is crucial. However, existing methods either fall short in maintaining the overall structure or exist intersections causing artifacts or distortions. In this work, we propose a vector-based Procrustes loss, and add positional relationship constraint to mitigate these two problems respectively.

**Vector-based Procrustes loss.** Although SDS can provide strong semantic guidance, the generated glyphs are prone to overfit the target shape, compromising the essential geometric structure of the original glyph and thus impairing legibility. To address this issue, we introduce Procrustes distance (Wang & Mahadevan, 2008), a metric that quantifies the dissimilarity between two sets of corresponding points by optimally aligning them through geometric transformations, thus preserving overall structural coherence in SVG glyph representation. Mathematically, given two sets of corresponding points $X = [x_1, x_2, \ldots, x_n]$ and $Y = [y_1, y_2, \ldots, y_n]$, the Procrustes distance is the minimum root-mean-square (RMS) distance between them after aligning one set to the other via translation, scaling, and rotation, defined as

$$D_{\text{proc}}(X, Y) = \min_{R, t, s} \|sXR + t - Y\|_F, \qquad (6)$$

where R is an orthogonal rotation matrix, t is a translation vector, s is a scaling factor, $\|\cdot\|_F$ denotes the Frobenius norm. As presented in Fig.3, our approach operates at the path level to maintain the glyph's topology. First, the outlines of $C_{letter}$ are extracted and decomposed into multiple paths, each comprising Bézier curves defined by control points. During optimization, we apply the Procrustes distance between each path of the intermediate glyph $x_{o_n}$ and its counterpart in the initial vector glyph $x_{o_0}$. Due to limited control points available in the SVG representation, we take a forward step: sampling more control points along each

path to enforce stronger structural constraints. Formally, the Procrustes loss is given as

$$\mathcal{L}_{Procrustes} = \sum_{i=1}^{N_p} ProDist(CP(x_{o_n,i}), CP(x_{o_0,i})). \qquad (7)$$

Here, $N_p$ represents the number of outline paths, $x_{o_n,i}$ denotes the $i^{th}$ path of the glyph $x_{o_n}$ at step $o_n$, and $CP$ refers to the set of all control points along these paths. This loss function thus maintains structural consistency of the generated glyphs by geometric transformation and strict path-level alignment with the original characters.

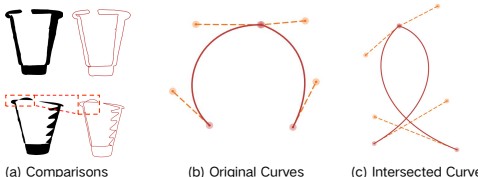

(a) Comparisons     (b) Original Curves     (c) Intersected Curves

*Figure 4.* (a) Generated results and their curves with/without (Top/Bottom) Positional Relationship Constraints. The red lines represent the intersection and its corresponding curves. (b) Original curves without intersections. Red/orange points denotes control points. (c) Artifacts often arise when two curves interset.

**Positional Relationship Constraints.** Bézier curves intersections often occur during SVG generation and lead to artifacts or improper topology, yet they are overlooked in Semantic Typography. As shown in Fig.4 (b) and (c), we observe that intersections tend to arise in a vector glyph when the original control points lose their clockwise order during optimization, causing two curves to intersect.

We assume that unconstrained optimization of control points leads to curve crossovers. To suppress this problem, we propose Positional Relationship Constraints with two key criteria: (1) maintaining the absolute ordering of control points, and (2) preventing excessively close placement between non-adjacent points. An example in Fig.4(a) shows the effectiveness of the proposed method. Formally, given a closed path $P_i$ composed of $N_t = N_c \times N_d$ control points, we construct $N_t$ line segments to ensure closure. For each control point, we identify its nearest neighbor and measure the Euclidean distance $d_k$. A learnable threshold $\delta$, which can dynamically adapt based on glyph's scale and complexity, is employed to detect pairs of control points that are too close. If such a pair is not adjacent in the original glyph structure, their proximity is penalized with a weight of $\lambda_s$. To enable efficient nearest neighbor queries, we index the 2D control point coordinates using a KDTree (Bentley, 1975). The intersection loss is defined as:

$$\mathcal{L}_{\text{inter}} = \lambda_s \sum_{(i,j) \in \mathcal{P}} \text{ReLU}(\delta - d_{ij}), \qquad (8)$$

where $\mathcal{P}$ denotes non-adjacent points set. This constraint encourages control points to retain appropriate spatial rela-

*Table 1.* Quantitative results of vector-based and raster-based models. Bold indicates the best result. $^*$ denotes $p < 0.05$ by two-tailed t-test between Word-As-Image and VecDesigner.

| Method | OCR(%)↑ | CD ↓ | CLIPScore↑ | BLIPScore↑ |
|---|---|---|---|---|
| *Raster-based Methods* | | | | |
| DS-Fusion | 79.23 | - | 0.2445 | 0.3383 |
| MetaDesigner | 81.72 | - | 0.2652 | 0.3416 |
| DaLL·E3 | 83.58 | - | 0.2685 | 0.3534 |
| Nano Banana Pro | 83.71 | - | **0.2739** | **0.3554** |
| *Vector-based Methods* | | | | |
| CLIPDraw | 72.65 | 1.6873 | 0.2467 | 0.3212 |
| Word-As-Image | 82.36 | 1.4922 | 0.2518 | 0.3238 |
| **VecDesigner** | **85.36** | **1.3717**$^*$ | 0.2733$^*$ | 0.3535$^*$ |

tionships and ordering, thereby reducing the risk of curve intersections during optimization.

### 4.4. Optimization Objective

Our final objective is to minimize the following term

$$\min_{\theta} \nabla_{\theta}\mathcal{L}_{\text{visual}} + \lambda_p \mathcal{L}_{Procrustes} + \mathcal{L}_{inter}, \qquad (9)$$

where $\lambda_p$ controls the strength of structural constraints.

## 5. Experiments

### 5.1. Experimental Setup

**Dataset.** We collect 100 concept prompts across five categories: animals, scenes, natural elements, food, and objects, with each category comprising 20 concepts. We also select five diverse font styles to evaluate our method across varying typographic characteristics.

**Baselines.** We compare our method with vector-based methods, including CLIPDraw (Frans et al., 2022), Word-As-Image (Iluz et al., 2023), and raster-based methods, including DS-Fusion (Tanveer et al., 2023), MetaDesigner (He et al., 2025), DaLL·E3 (Betker et al., 2023) and Nano Banana Pro (Team et al., 2023).

**Evaluation Metrics.** Evaluating semantic typography is challenging due to the absence of a clear ground truth. To address this, we focus on two essential aspects: semantic alignment with the input text and character legibility. Semantic alignment is quantified using CLIPScore (Radford et al., 2021), BLIPScore (Li et al., 2022) with prompts formatted as "Picture of <concept word>". For character legibility, we leverage GPT-4o's OCR (Hurst et al., 2024) to measure whether the generated glyphs are accurately recognized as the original ones, and we also employ bidirectional Chamfer distance between the initial and generated control points to evaluate their structural similarity.

**Implementation Details.** Our framework is built on the pre-trained Stable Diffusion 2-1-base model through PyTorch (Paszke et al., 2019), with all hyperparameters kept to their default values. The optimization process is conducted over 500 iterations. The hyperparameters are set as follows: $\lambda_v = 1.0$, $\lambda_p = 0.01$, $\lambda_s = 100$, $N_s = 5$, with 10 additional sampling points added along each path. The reference images are generated every 500 steps using a guidance scale of 7.5. More details can be found in Appendix.

### 5.2. Comparison with Existing Methods

**Quantitative Results.** As shown in Table 1, VecDesigner achieves state-of-the-art performance across all metrics. For semantic representation, VecDesigner achieves 0.2733 in CLIPScore and 0.3535 in BLIPScore, much higher than previous vector-based method, Word-As-Image. For character legibility, VecDesigner attains a 3% improvement in OCR accuracy, which serves a coarse-grained evaluation. For a more granular assessment, it achieves a 0.1205 reduction in Chamfer Distance. We also perform a two-tailed t-test on Chamfer Distance, CLIPScore and BLIPScore between VecDesigner and Word-As-Image. The results indicate that these improvements are statistically significant ($p < 0.05$). The performance gains reflect that VecDesigner exhibits superior semantic alignment with target meaning and enhanced legibility compared to other methods. Notably, VecDesigner exceeds almost all raster-based approaches, further demonstrating its effectiveness in semantic expressiveness and structural integrity. We also conduct user studies to evaluate the quality of vector-based results. Details and results are presented in Appendix.

**Qualitative Results.** Fig.5 showcases representative outputs from all evaluated approaches. CLIPDraw frequently fails to integrate semantic concepts with the structural integrity of the characters. Word-as-Image occasionally produces ambiguous results—for example, the second-row output resembles a ghost rather than a monkey. In cases like "COW" and "ARCH," the generated results lack both semantic clarity and structural coherence. Even when the concept is captured, the underlying glyph shape is often compromised, as shown in the deformation of "F" into a leaf-like shape. Among pixel-based approaches, DS-Fusion falls short in balancing semantic meaning and character structure, while DALL·E3 struggles to reshape glyphs in a semantically meaningful way. Although MetaDesigner successfully maintains glyph structural integrity, it fails to convey semantic content in an aesthetically coherent and visually compelling manner. Nano Banana Pro can generate vivid and clear glyphs, though sometimes at the expense of legibility. In contrast, our approach preserves character legibility while effectively conveying the intended semantic concepts, and further enhances visual expressiveness and vividness in the generated illustrations.

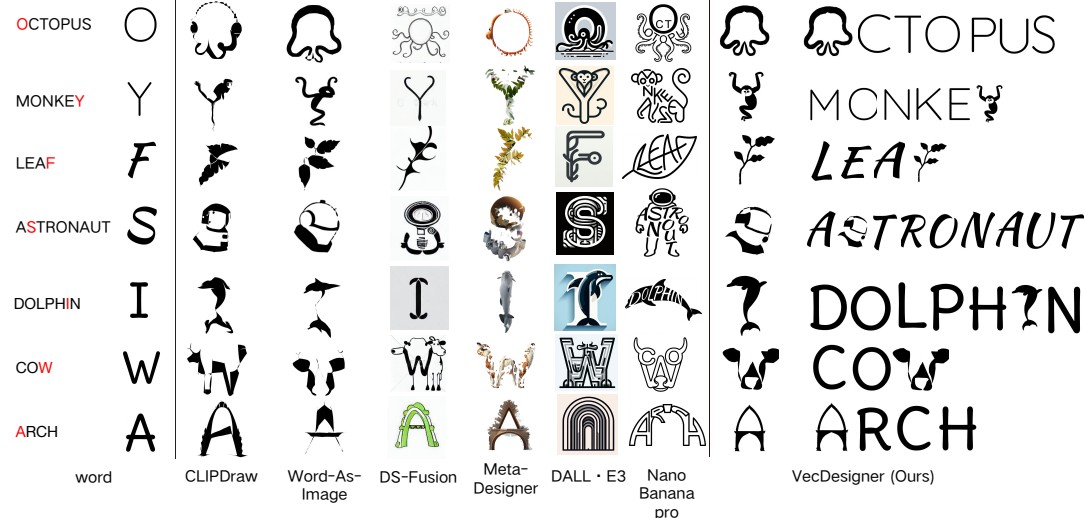

*Figure 5.* Visual comparisons of competing vector-based and raster-based methods on semantic typography.

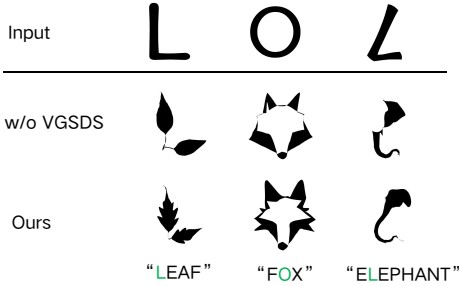

*Figure 6.* Visualization of ablation on VGSDS. Our method excels at generating glyphs with rich semantic details.

*Table 2.* Ablation Studies of VecDesigner. VG, $\mathcal{L}_P$ and $\mathcal{L}_i$ represent for VGSDS, $\mathcal{L}_{Procrustes}$ and $\mathcal{L}_{iter}$ respectively.

| Components | | | OCR(%) ↑ | CD ↓ | CLIPScore ↑ | BLIPScore ↑ |
|---|---|---|---|---|---|---|
| VG | $\mathcal{L}_P$ | $\mathcal{L}_i$ | | | | |
| ✗ | ✗ | ✗ | 81.26 | 1.4786 | 0.2640 | 0.3419 |
| ✔ | ✗ | ✗ | 83.55 | 1.4816 | 0.2646 | 0.3402 |
| ✔ | ✔ | ✗ | 84.18 | 1.4495 | 0.2700 | 0.3421 |
| ✗ | ✔ | ✔ | 84.33 | 1.4570 | 0.2714 | 0.3428 |
| ✔ | ✗ | ✔ | 84.13 | 1.4556 | 0.2720 | 0.3467 |
| ✔ | ✔ | ✔ | **85.36** | **1.3717** | **0.2733** | **0.3535** |

*Table 3.* Performance of reference images generation intervals.

| Intervals | OCR(%) ↑ | CD ↓ | CLIPScore ↑ | BLIPScore ↑ |
|---|---|---|---|---|
| 50 | 83.21 | 1.4627 | 0.2703 | 0.3460 |
| 100 | 83.30 | 1.4455 | 0.2682 | 0.3427 |
| 250 | 85.07 | 1.4594 | 0.2710 | 0.3440 |
| **500** | **85.36** | **1.3717** | **0.2733** | **0.3535** |

## 5.3. Ablation Study

**Influence of Each Component.** The ablation baseline is vanilla SDS framework without any constraints. We separate the proposed techniques and sequentially add them to the baseline. The quantitative results can be seen in Table 2, verifying that VGSDS, $\mathcal{L}_{Procrustes}$ and $\mathcal{L}_{inter}$ all can help improve the quality of generated glyphs. Moreover, we provide visual comparisons of the ablation study results. As depicted in Fig.6, when VGSDS is removed, the visual quality of glyphs deteriorates. It indicates that visual guidance contributes to enhancing semantic details. As illustrated in Fig.8, adding $\mathcal{L}_{Procrustes}$ makes the glyphs more structurally coherent and legible. As shown in Fig. 9, the red boxes highlight intersections in generated glyphs and their corresponding curves. The integration of $\mathcal{L}_{inter}$ alleviates artifacts of intersections to a large extent.

**Influence of Interval of Semantic Images Generation.** We conduct experiments with varying intervals for using the pre-trained diffusion model to generate reference images.

By default, this interval is set to 500 optimization steps. As presented in Table 3, updating references at lower frequencies leads to improved overall performance. It implies that infrequent updates helps maintain less noisy, more stable and meaningful semantic guidance during optimization.

**Influence of Numbers of Sampling Points.** As shown in Table 4, we experiment with various numbers of additional sampling points. The performance indicates that introducing a moderate number of additional points enhances both semantic alignment and legibility, while excessive sampling may hinder optimization. This phenomenon occurs because

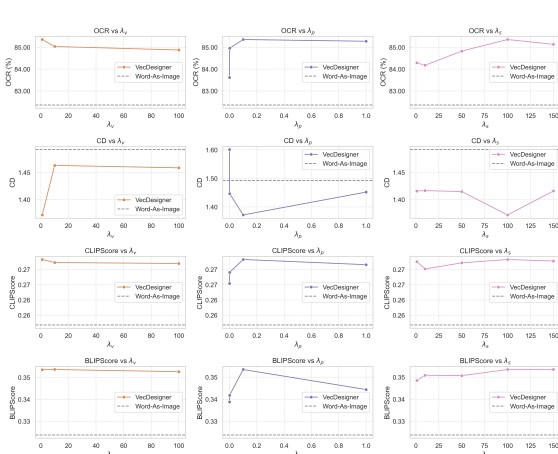

*Figure 7.* Visualization on various fonts and concepts. VecDesigner can generate high-quality glyphs for distinct font styles.

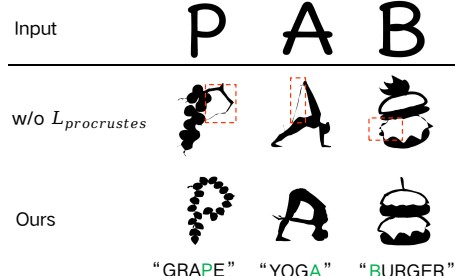

*Figure 8.* Visualization of ablation on $\mathcal{L}_{procrustes}$. The regions highlighted by the red boxes indicate areas of structural degradation in the generated glyphs, leading to poor legibility.

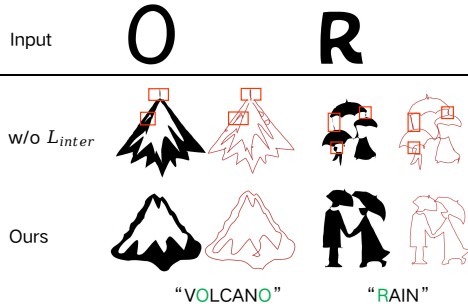

*Figure 9.* Visualization of ablation on $\mathcal{L}_{inter}$. The red boxes annotate the flawed regions in the rendered glyphs and pinpoint their corresponding locations on the Bézier curves of the character.

more points provide stricter constraints in overall structure, yet excessive sampling introduces redundancy and noise. During optimization, the algorithm is prone to mistakenly take these noises as key features of the curve for fitting, which instead deviates from the true shape of the curve.

**Parameter Sensitivity.** The impact of varying $\lambda_v$, $\lambda_p$ and $\lambda_s$ are shown in Fig.10. The performance are consistently stable across all evaluations, demonstrating the robustness of our proposed framework.

*Figure 10.* Impact of hyperparameters on model performance.

## 5.4. Visualization

Fig. 7 illustrates more examples with various characters and fonts. The Appendix contains further visualizations.

*Table 4.* Performance of varing numbers of additional sampling points along each path in Procrustes loss.

| Number | OCR(%) ↑ | CD ↓ | CLIPScore ↑ | BLIPScore ↑ |
|--------|----------|--------|-------------|-------------|
| 0 | 84.08 | 1.4418 | 0.2686 | 0.3418 |
| 5 | 84.74 | 1.4590 | 0.2710 | 0.3438 |
| **10** | **85.36** | **1.3717** | **0.2733** | **0.3535** |
| 20 | 84.90 | 1.4637 | 0.2688 | 0.3405 |
| 30 | 85.26 | 1.4586 | 0.2702 | 0.3454 |

## 6. Conclusion

This paper proposes VecDesigner, a novel framework for vector semantic typography. We address the limitations of existing methods by introducing an extension of SDS-VGSDS, that leverages visual guidance to enrich semantic details. We introduce two structural constraints to maintain

glyph legibility and crossovers artifacts. Extensive experimental results demonstrate that VecDesigner achieves in both meaningful visual interpretation and structural coherence. Limitations are further discussed in Appendix.

## Acknowledgements

This work was supported by the National Nature Science Foundation of China (62406073), the Fundamental Research Funds for the Central Universities, and the computation is performed in ECNU Multifunctional Platform for Innovation(001).

## Impact Statement

This paper presents work whose goal is to advance the field of vector-based Semantic Typography. We propose VecDesigner to improve the effectiveness of generating semantically meaningful, visually expressive, and structurally coherent glyphs. These advances may benefit digital communication, creative industries, and accessible visual design by enabling clearer and more expressive typographic representations.

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

# A. Related Work

## A.1. Raster-based Sematic Typography

Semantic Typography requires transforming text shapes to visually express specific concepts or objects. Raster-based methods operate directly on pixel-level images. Early approaches attempt to identify images that match the semantics of words and intricately combine them according to the structure of glyph strokes (Freepik, 2010). However, this manual process is often laborious and time-consuming for artists. To address this challenge, Zhang (Zhang et al., 2017) designs an interactive system that enables users to segment glyphs into several strokes. It incorporates a semantic-shape similarity metric to match image shapes with corresponding strokes. Trick and TReAT (Tendulkar et al., 2019) introduces a more automatic approach by training an autoencoder to map both clipart and letter images in the same latent space, allowing for the measurement of their visual similarity. The system further reconstructs the clipart that retains the structural characteristics of the original letter. Recently, large pre-trained generative models (Rombach et al., 2022) have demonstrated their ability in semantic typography. DS-Fusion (Tanveer et al., 2023) is a diffusion-based model with an additional CNN-based discriminator, which processes characters in raster format, focusing on their stylistic features and textural details. VitaGlyph (Feng et al., 2024) segments a glyph into subject and surrounding regions, and employs a dual-branch ControlNet (Zhang et al., 2023) model that separately processes each region to synthesize the complete text. WordArt Designer (He et al., 2023) utilizes the capabilities of Large Language Models (LLMs) (Achiam et al., 2023) for artistic typography. MetaDesigner (He et al., 2025) introduces a multi-agent framework, integrating evaluation and optimization modules to enhance the creation of artistic text styles. FonTS (Shi et al., 2025) proposes a novel DiT-based pipeline for controllable typography and style in text rendering. Although pixel-based images can capture complex textures and contain richer color information, they suffer from inherent drawbacks such as resolution dependency and tightly coupled with the background, lacking editability.

# B. Implementation Details

We employ the Adam optimizer (Kingma & Ba, 2014) with parameters $\beta_1 = 0.9, \beta_2 = 0.9, \epsilon = 1e^{-6}$. The optimization process is conducted over 500 iterations. The learning rate is linearly increased from $2e^{-4}$ to $2e^{-3}$ during the initial 100 warm-up steps, followed by a decay to $8e^{-4}$ over the remaining 400 iterations. All experiments are conducted on RTX 4090. We adopt the prompt format used in Word-As-Image, using the following prompt for each target concept word: "[word]. minimal flat 2d vector. lineal color. trending on artstation.". All the input images are resized

*Table 5.* Ablation study on data augmentation. We employ the same data augmentation strategy with previous state-of-the-art models and outperform them even without augmenting data.

| Method | OCR (%) ↑ | CD ↓ | CLIPScore ↑ | BLIPScore ↑ |
|---|---|---|---|---|
| CLIPDraw | 72.84 | 1.6873 | 0.2411 | 0.3196 |
| Word-As-Image | 82.09 | 1.4922 | 0.2472 | 0.3174 |
| w/o augmentation | 83.47 | 1.3949 | 0.2718 | 0.3448 |
| **VecDesigner** | **85.36** | **1.3717** | **0.2733** | **0.3535** |

*Table 6.* Ablation study on initial control points. "Default Number" means the number of control points inherent set in FreeType Font Library.

| Method | OCR(%)↑ | CD↓ | CLIPScore↑ | BLIPScore↑ |
|---|---|---|---|---|
| Default Number | 85.11 | 1.4335 | 0.2678 | 0.3426 |
| **VecDesigner** | **85.36** | **1.3717** | **0.2733** | **0.3535** |

to $512 \times 512$. Following prior works such as CLIPDraw and Word-As-Image, we also apply augmentation to the rasterized image before feeding it to the diffusion model at each iteration. Our method is applied individually to each character in the concept word.

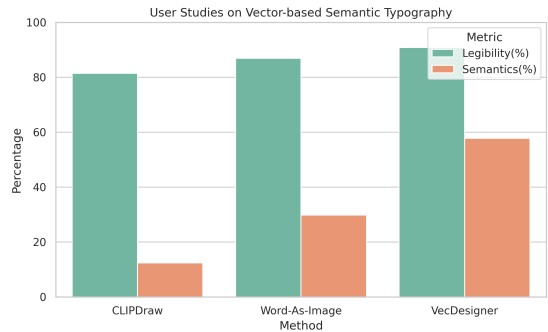

*Figure 11.* The results of user study on vector-based Semantic Typography.

# C. Details of User Study

We conduct user studies to evaluate the quality of vector-based results. For each of the five fonts in our dataset, five concept words were randomly selected, resulting in a total of 25 test cases. We recruit 20 participants with diverse academic and professional backgrounds to assess the results. To evaluate legibility, users were asked to identify the intended character represented by each generated glyph. For semantic accuracy, they were instructed to select a result that best matched the given concept. As shown in Fig.11, VecDesigner achieves the highest identification accuracy among all methods, and a substantially larger proportion of participants judge its outputs as more semantically representative than other methods.

*Table 7.* Quantitative results on Chinese dataset.

| Method | OCR(%)↑ | CD ↓ | CLIPScore↑ | BLIPScore↑ |
|---|---|---|---|---|
| CLIPDraw | 83.50 | 1.6092 | 0.2149 | 0.2641 |
| Word-As-Image | 84.73 | 1.4582 | 0.2490 | 0.3094 |
| **VecDesigner** | **85.26** | **1.3523** | **0.2739** | **0.3603** |

## D. More Ablation Studies

### D.1. Influence of Data Augmentation

We investigate the impact of applying data augmentation to raster images prior to input into the pre-trained Stable Diffusion model. As shown in Table 5, our method surpasses CLIPDraw and Word-As-Image significantly even without augmentation, highlighting its effectiveness.

### D.2. Influence of Initial Control Points

We employ the FreeType Font Library to extract character outlines defined by control points. The number of control points varies with font style and character complexity. Following Word-As-Image, we set a threshold for each character and recursively subdivide Bézier segments until this threshold is met. To assess the effect of initialization, we examine the impact of varying the number of control points. As shown in Table 6, a higher number of initial control points enhances geometric detail, enabling richer and more expressive visual representations.

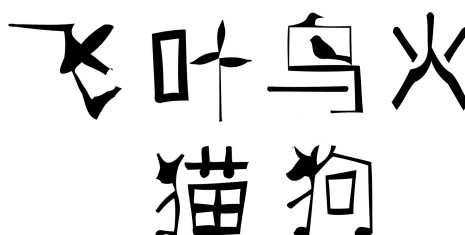

*Figure 12.* More visualization results produced by VecDesigner on Chinese fonts and characters.

## E. Non-Latin Evaluation

We also evaluate the methods on Chinese characters, including 4 Chinese fonts, with 20 concepts, 80 diverse letters in total. As shown in Table.7, VecDesigner outperforms others across all metrics, highlighting the cross-lingual ability of our method. Furthermore, the visualization of Chinese results are shown in Fig.12, verifying the generalization of our proposed method.

## F. Visualization

In this section, we present additional visualizations generated by VecDesigner. As illustrated in Fig.13, the results span a wide range of fonts and lexical content from our dataset. These visual examples demonstrate that the outputs produced by our method exhibit rich semantic detail and maintain high legibility across diverse font styles.

## G. Applications of VecDesigner

In this section, we explore downstream applications of VecDesigner. By using our generated results to support both Depth-to-Image and Scribble-to-Image pipelines, we provide strong structural and semantic priors that assist these models in producing richer and more detailed pixel-level images. This demonstrates the practical value of our task: the outputs of VecDesigner not only serve as high-quality visual representations on their own but also enhance the fidelity, diversity, and controllability of subsequent generative processes.

### G.1. Scribbles-to-Image Generation

In this subsection, we utilize our produced results as conditional input to ControlNet-SDXL to guide Scribbles-to-Image generation. As shown in Fig.15 , the results demonstrate that, the clean and semantically meaningful scribbles generated by VecDesigner facilitate ControlNet to generate images with clear geometry and enhanced visual fidelity.

### G.2. Depth-to-Image Generation

As shown in Fig.16, we utilize our produced results as conditional input to ControlNet-SDXL to guide Depth-to-Image generation. The results demonstrate that, on top of our depth maps, the models can generate tridimensional objects with diverse colors and rich textures, leading to images that exhibit both stronger spatial realism and enhanced visual richness.

## H. Limitations and Future Work

Our method has two limitations. First, it struggles to effectively convey semantic concepts whose visual representations significantly deviate from the inherent structure of the glyph. This challenge arises because not all characters are suitable for deformation into specific objects. For example, as shown in the first row in Fig.14, the letters "D" and "R" are not suitable to represent the form of a deer. Moreover, while each letter in "SOCK" resembles a sock, their legibility is significantly compromised. To mitigate this issue, we suggest developing a specific algorithm to match the shapes of semantic concepts with compatible characters, thereby enabling more efficient and meaningful generation, rather

FOX DESERT SHARK
TEMPLE CHERRY FROG
DOG CUP SNAKE HAT
BEAR GRAPE ROBOT
KEY PALACE HAT OWL
DONUT WRENCH STAR
CAKE CAT TIGER BAG
CLOUD KNIFE COFFEE
GINGER CRSTAY SINGER

*Figure 13.* The results of textual studies on vector-based semantic typography methods.

than uniformly applying the method to all glyphs. Secondly, our method requires improvement in holistically deform an entire semantic concept, which is also a challenging problem for current methods. For instance in the second row in Fig.14, the word "TREE" cannot be effectively deformed into a tree shape. To address this, we split it into "TR" and "EE", aligning each with a tree component. However, integrating the two parts into a coherent whole presents a new challenge. A potential direction for improvement is to introduce LLMs to support the processing and deformation of complete words in a semantically coherent manner.

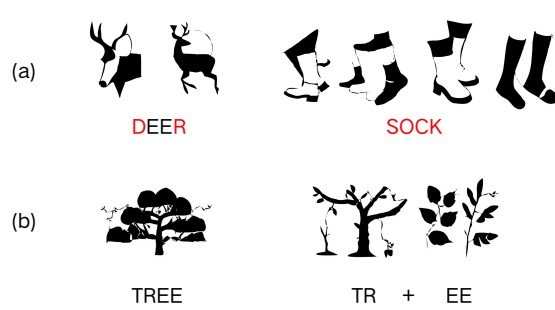

*Figure 14.* Failure cases illustration.

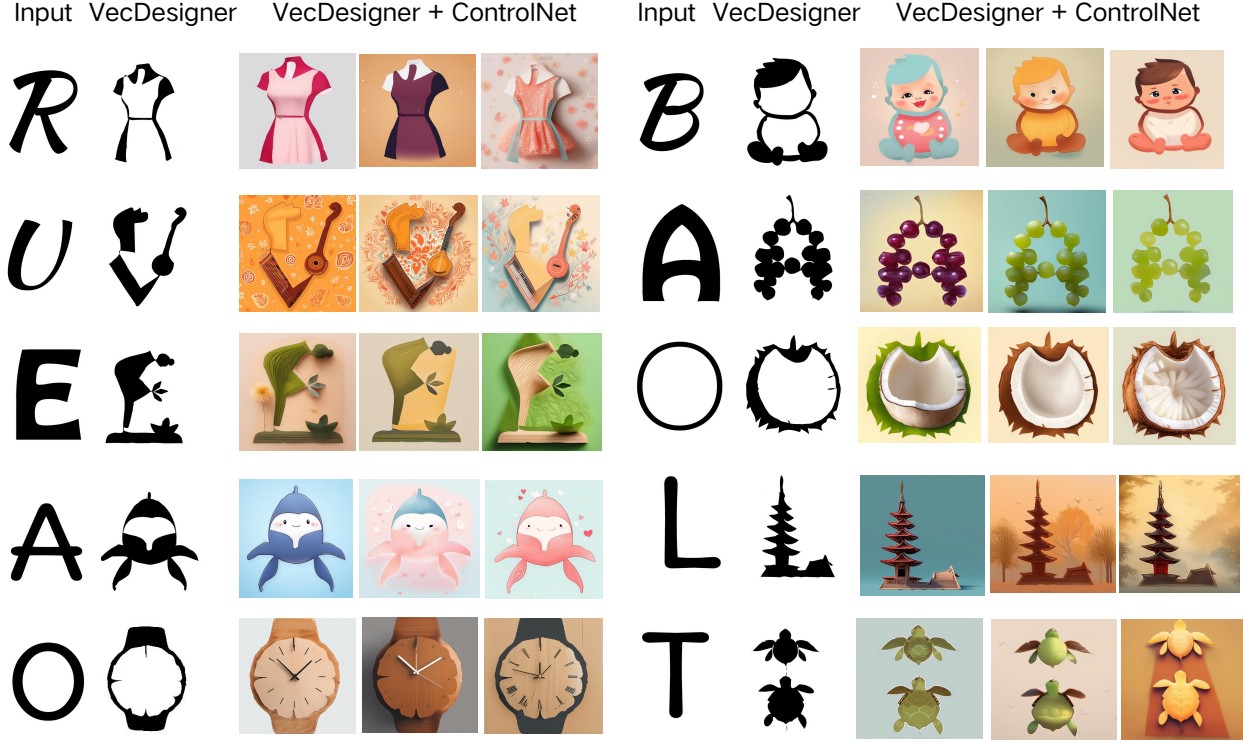

*Figure 15.* Examples of serving as condition of Scribbles-to-Image with ControlNet-SDXL.

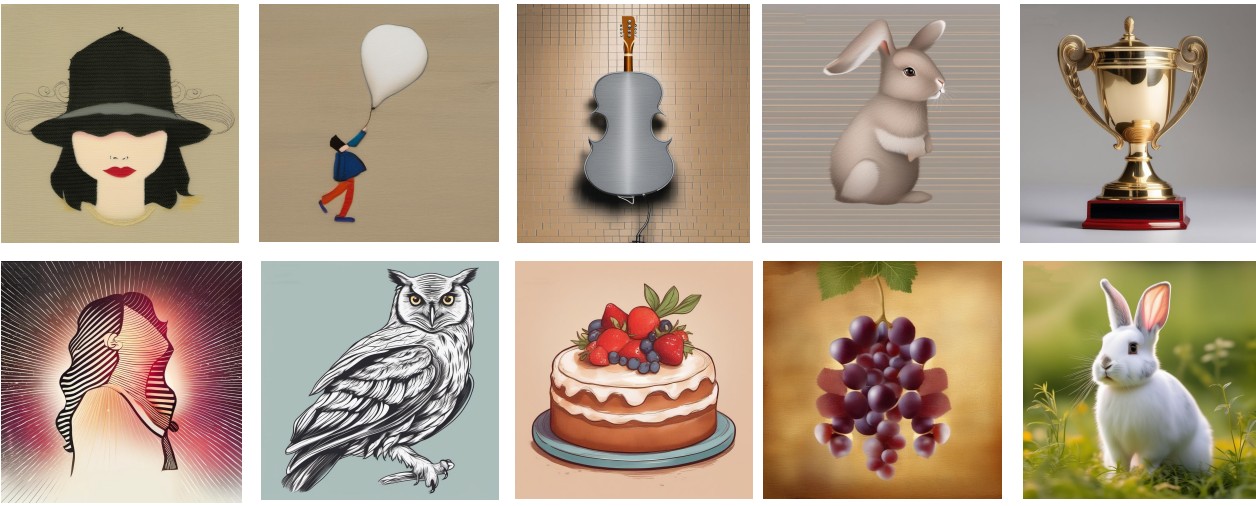

*Figure 16.* Examples of serving as condition of Depth-to-Image with ControlNet-SDXL.

