# OpenReview forum: "VecDesigner: Exploring Visual Guidance and Structural Consistency for Semantic Typography"
_ICML.cc/2026/Conference — ICML 2026 regular_

### Official Review · Reviewer_DjkT · 2026-03-09

**Soundness:** 3
**Presentation:** 3
**Significance:** 2
**Originality:** 3
**Overall Recommendation:** 4
**Confidence:** 4

**Summary:**

This paper explores vector-based typography generation using score distillation sampling (SDS) with a pre-trained text-to-image model. Compared with the vanilla baseline, the authors propose several regularization method to improve the quality of the generated typography. First, they propose to use an additional loss based on the CLIP image embedding of the intermediate glyph and the reference image to improve the expressiveness. Second, a vector-based Procrustes loss and an intersection loss as structural constraints to help maintain glyph legibility and reduce the crossovers artifact.

**Compliance With Llm Reviewing Policy:**

Affirmed.

**Key Questions For Authors:**

I don't have additional questions.

**Limitations:**

Yes

**Strengths And Weaknesses:**

[Soundness]

The proposed method is well-motivated and quantitative experiments show that it outperform the existing raster-based and vector-based methods. Extensive ablation study is done to verify each component and help understand the functionality of different losses.

[Presentation]

The paper is very well-written and easy to understand.

[Significance]

The paper focuses on an incremental improvement of using SDS-based method for typography and proposes several losses to improve generation quality as well as reducing the generation artifacts. Though simple, these developments could be useful for future application of such methods. My only concern is whether it fits the ICML venue well.

[Originality]

The proposed method is novel based the best of my knowledge.

---

> ### Author Rebuttal · Authors · 2026-03-31
>
> **R1 - Venue concern:** We sincerely appreciate the time and effort you have dedicated to reviewing our manuscript. We understand the concern about the venue fit, so we looked for papers with closely related research topics in SVG generation published on ICML in recent years. For example, [1] proposes a text-guided SVG generative model that can draw vector graphics by predicting individual shapes and positions. [2] introduces StrokeNUWA, which designs a novel type of stroke token for LLMs to better synthesize vector graphics. Motivated by these precedents, we believe our work aligns well with the scope of ICML and contributes to the development of this field.
>
> [1]Cipriano, Marco, Moritz Feuerpfeil, and Gerard De Melo. "Vector Grimoire: Codebook-based Shape Generation under Raster Image Supervision." Forty-second International Conference on Machine Learning. 2025.
> [2]Tang, Zecheng, et al. "StrokeNUWA: tokenizing strokes for vector graphic synthesis." Proceedings of the 41st International Conference on Machine Learning. 2024.

---

> > ### Author Rebuttal · Reviewer_DjkT · 2026-04-04
> >
> > I will keep my rating.

---

> > > ### Author Response · Authors · 2026-04-04
> > >
> > > Thank you again for your encouraging review and support.

---

### Official Review · Reviewer_D5eA · 2026-03-13

**Soundness:** 2
**Presentation:** 3
**Significance:** 3
**Originality:** 2
**Overall Recommendation:** 4
**Confidence:** 3

**Summary:**

This paper focuses on the task of Semantic Typography. It analyzes two primary issues with existing vector generation methods that typically rely on pure text-driven optimization techniques (such as diffusion model-based Score Distillation Sampling, or SDS): the generated fonts lack rich semantic details, and their structures are highly prone to collapse. To tackle these challenges, the authors propose a novel optimization-based method called VecDesigner. Specifically, the paper introduces Visual-Guided Score Distillation Sampling (VGSDS) to address the issue of insufficient details. Furthermore, a Vector-based Procrustes loss is designed to resolve the structural collapse problem and preserve legibility, while Positional Relationship Constraints are applied to eliminate intersection artifacts. Finally, the paper conducts comprehensive experiments to verify the effectiveness of the proposed method.

**Compliance With Llm Reviewing Policy:**

Affirmed.

**Final Justification:**

My concerns are resolved. Thank you. I have changed my rating to WA.

**Key Questions For Authors:**

1. What is the comparative computational cost and inference time of VecDesigner?
2. Beyond the few provided samples, how does the proposed framework generalize to non-English and structurally complex characters compared to existing baseline methods?
3. How to address the inherent contradiction between the Procrustes Loss and semantic deformation? Since the Procrustes distance intrinsically only permits rigid/affine transformations (i.e., translation, scaling, and rotation), if the topological structure of the semantic target (e.g., a "Deer") is completely incompatible with the base letter (e.g., "D"), the rigid Procrustes constraint will lead the optimization into a deadlock. Does this suggest a fundamental design flaw in the application of the Procrustes Loss for this task?
4. Regarding the penalty mechanism of the positional constraint, how does the framework avoid preventing the formation of reasonable high-frequency details or sharp inner corners? If the semantic target requires the letter to develop fine "hairs" or "spikes", the current loss function would likely mistakenly penalize these features as "intersection risks" and inappropriately smooth them out.

**Limitations:**

yes

**Strengths And Weaknesses:**

**Strengths**

The paper thoroughly analyzes the limitations and issues of existing methods, establishing a solid and well-justified motivation.

The authors propose three targeted components to specifically address these identified issues. These components offer valuable insights and inspiration for broader vector-based image generation tasks.

**Weaknesses**

1. The experiments on characters in languages other than English are limited.

2. There is a lack of comparison regarding inference/testing time. Since the proposed approach is an optimization-based method utilizing SDS, a computational speed comparison with existing feed-forward methods is missing.  The authors criticize data-driven methods for being "resource-intensive". However, VecDesigner, as a purely optimization-based method utilizing SDS, requires 500 complete forward passes of the diffusion model and backward passes of the DiffVG renderer for the generation of each individual letter. The computational overhead of such test-time optimization is exceedingly high, making it entirely incapable of meeting the real-time interaction requirements of practical typographic design.

3. The primary motivation for proposing VGSDS is the authors' assertion that glyphs generated by plain-text SDS are too abstract and lack visual details. However, the current evaluation is limited to character recognition (OCR) and image-level CLIP metrics, noticeably lacking any aesthetic assessments or subjective preference comparisons.

---

> ### Author Rebuttal · Authors · 2026-03-31
>
> Thank you for your constructive comments. We address your concerns one-by-one below.
>
> **Q1/W2 - Computational cost:** We conduct a comparison with previous vector-based Semantic Typography methods on an NVIDIA RTX 4090. To ensure fairness, we measure the average inference time per image across five randomly-selected concepts using all fonts for each category in the test set. We follow the default experimental settings of each method. Table-Q1/W2 provides the inference time comparison results.
>
> **Table-Q1/W2 Comparison of Inference time**
> |Methods|Steps|Inference time|
> |-|-|-|
> |CLIPDraw|1000|2m32s|
> |Word-As-Image|500|3m32s|
> |**Ours**|500|3m57s|
>
>
> **Q2/W1 - Non-English & Complex characters:** Thanks for your advice. Firstly, compared with vanilla SDS, VGSDS further aligns intermediate character features with reference images. This design is language-agnostic and can be applied to other non-Latin languages. Secondly, since our vector-based Procrustes loss operates at the path level and samples more points along the paths, it can reinforce the proper deformation of every stroke of more complex characters. Moreover, the dense contours in intricate or some non-Latin characters are more prone to intersect. The proposed Positional Relationship Constraints utilize a KDTree for neighbor queries and apply a dynamically adaptable threshold $\delta$, which can adapt to the glyph’s complexity. To assess generalization, we have conducted comparisons on a Chinese dataset, as shown in Table 7 in Appendix. VecDesigner achieves superior performance on both semantic and structural metrics, demonstrating its effectiveness.
>
>
> **Q3 - Clarification of Procrustes loss and semantic deformation:** We sincerely thank the reviewer for raising this critical question. We want to clarify that this is not a design flaw, but an essential regularization mechanism. The objective of Semantic Typography is to achieve a trade-off between semantic expressiveness and character readability. During optimization, VGSDS will push the control points to undergo non-rigid deformations to align with the semantics. Simultaneously, the vector-based Procrustes loss would penalize displacement that leads to structural collapse or over-deformation, instead of a strict constraint that forbids non-rigid deformation. Therefore, in our framework, this joint objective allows the model to find a boundary where the character can warp into the semantic target.
>
>
> **Q4 - Clarification of positional constraint:** We appreciate the reviewer for raising this question. The Positional Relationship Constraint is designed to penalize the disordering of Bézier control points to prevent visual artifacts. When a semantic target requires reasonable high-frequency details or sharp inner corners, the control points move collectively to form target curvature. As long as these curves remain regular and ordered, the constraint does not penalize them or smooth them out. Conversely, disordered control points result in jagged contours and intersections, severely degrading character legibility.
> For example, in the ''RAIN'' example in Fig.9, the umbrella generated by VecDesigner exhibits sharp corners while maintaining coherent contours; without this constraint, these clean inner corners would degrade into tangled, overlapping regions.
>
>
> **W3 - Evaluation:** Thank you for your suggestion. For the aesthetic assessment, we leverage ArtiMuse [1], an MLLM-based image aesthetic assessment (IAA) model with scoring and semantic understanding capabilities, to evaluate the performance of vector-based methods. For each method, We randomly choose 20 results with the same fonts and concepts, and compute the average scores across the eight fine-grained aesthetic scores proposed by ArtiMuse. The results are presented in Tale-W3. It is clear to see that our method achieves the highest score in the comprehensive evaluation.
>
> **Table-W3 Comparison of aesthetics assessment**
> |Model|Composition & Design|Visual Elements & Structure|Technical Execution|Originality & Creativity |Theme & Communication|Emotion & Viewer Response|Overall Gestalt|Comprehensive Evaluation|
> |-|-|-|-|-|-|-|-|-|
> |CLIPDraw|5.58|2.08|7.45|2.06|6.06|5.71|3.77|4.47|
> |Word-As-Image|7.29|2.79|8.22|3.03|7.03|4.55|2.68|4.73|
> |**Ours**|7.13|3.63|8.05|3.01|6.81|5.21|6.02|5.65|
>
> For subjective preference comparisons, we have conducted a user study as described in the main text (Line 303-306) , with detailed results presented in Appendix C: Details of User Study.
>
> [1]Cao, Shuo, et al. "Artimuse: Fine-grained image aesthetics assessment with joint scoring and expert-level understanding." arXiv preprint arXiv:2507.14533 (2025).

---

> > ### Author Rebuttal · Reviewer_D5eA · 2026-04-04
> >
> > My concerns are well resolved. Thanks.

---

> > > ### Author Response · Authors · 2026-04-04
> > >
> > > Thank you again for your feedback and for helping us sharpen the clarity and integrity of the work.

---

### Official Review · Reviewer_puwx · 2026-03-14

**Soundness:** 3
**Presentation:** 3
**Significance:** 2
**Originality:** 3
**Overall Recommendation:** 4
**Confidence:** 4

**Summary:**

This paper proposes VecDesigner, a framework for vector semantic typography. To address the limitations of existing methods, the paper introduces an extension of SDS called Visual-Guided Score Distillation Sampling (VGSDS), which leverages generated visual guidance to enrich concrete visual details of the prompt. To preserve legibility and structural integrity, the framework employs a vector-based Procrustes loss. Furthermore, the intersection loss of Positional Relationship Constraints is incorporated to prevent undesired path intersections. Quantitative and qualitative experiments demonstrate that VecDesigner outperforms existing methods from both semantic and structural aspects.

**Compliance With Llm Reviewing Policy:**

Affirmed.

**Final Justification:**

My concerns are well resolved in the rebuttal. I will maintain my score.

**Key Questions For Authors:**

1. What is the specific reason for averaging the CLIP features of sampled images? When the images have high diversity, will this cause semantic dilution?
2. What is the computational resource required? How about the total optimization time?
3. How does the framework handle fonts where the letters are connected?

**Limitations:**

yes

**Strengths And Weaknesses:**

Strengths:
1. The paper proposes a creative method that incorporates VGSDS to introduce concrete semantic information into the optimization of vector typography. Additionally, the use of Procrustes distance is a smart metric choice for the readability constraint.
2. Extensive quantitative and qualitative experiments were conducted for a comprehensive evaluation of the framework. The results balance the semantic legibility and readability.
3. The presentation of the paper is good. Each component's rationale is clearly articulated. The illustrations are both intuitive and easy to follow.

Weaknesses:
1. The loss design may cause over-constraining. For example, in the "sock" case in Figure 14, the socks fail to be flexible enough to fit in the glyph shape. This suggests that the structural constraints may be too strict or there's a lack of flexibility in semantic shapes.
2. The potential semantic dilution or diversity loss in the average operation. The operation of averaging the CLIP features from sampled reference images may cause semantic dilution in the case of highly diverse data. Moreover, while increasing the number of sampled images improves the overall performance, it could result in a lack of diversity in the output glyph.

---

> ### Author Rebuttal · Authors · 2026-03-31
>
> Thank you for your constructive comments. We address your concerns one-by-one below.
>
> **Q1/W2 - Averaging CLIP features:** Thanks for pointing it out. Since all generated images are conditioned on the same text prompt, they inherently share the target semantic meaning. Averaging these diverse image features can filter out inessential visual differences, such as backgrounds or layouts, and reinforce core semantics. Therefore, when the images have high diversity, instead of semantic dilution, averaging CLIP features will prevent vector glyph from overfitting to the visual noise or a specific reference image.
>
> **Q2 - Computational Resource:** We conduct a comparison with previous vector-based Semantic Typography methods on an NVIDIA RTX 4090. To ensure fairness, we measure the average inference time per image across five randomly-selected concepts using all fonts for each category in the test set. We follow the default experimental settings of each method. Table-Q2 provides the inference time comparison results.
>
> **Table-Q2 Comparison of Inference time**
> |Methods|Steps|Inference time|
> |-|-|-|
> |CLIPDraw|1000|2m32s|
> |Word-As-Image|500|3m32s|
> |**Ours**|500|3m57s|
>
> **Q3 - Connected letters.:** We appreciate the reviewers for this insightful question. In this paper, we focus on the letters that are unconnected, and VecDesigner operates on individual letters. Under the settings that multiple letters are connected, it is crucial to maintain the connection between adjacent letters unbroken during semantic deformation. While our work excels at preserving the structure of individual characters, generating connected text requires extending our framework to model inter-character relationships. We have explored the preliminary idea in the context of holistic word deformation, but it needs deeper research (Line 609-799 in Appendix). We deeply appreciate this valuable suggestion, and we will incorporate a more detailed discussion regarding connected letters in Future Work section of the revised manuscript.
>
> **W1 - Over constraining:** Thanks. The aim of Semantic Typography is to achieve a balance between maintaining character legibility and expressing detailed semantic meanings. The "sock" case in Figure 14 presents an extreme scenario where the semantic object has a simple tube appearance that fundamentally conflicts with the target letter. If the vector-based Procrustes loss is relaxed, the optimization would tend to degenerate the letter into a mere illustration of a sock, severely compromising legibility. To address your concern about flexibility, adjusting the weight of the structural loss is a feasible approach, and  our sensitivity analysis has verified the robustness of the proposed method.

---

> > ### Author Rebuttal · Reviewer_puwx · 2026-04-01
> >
> > I will maintain my score.

---

> > > ### Author Response · Authors · 2026-04-04
> > >
> > > Thank you again for your encouraging review and support.

---

### Decision · Program_Chairs · 2026-04-30

**Decision:**

Accept (regular)

**Comment:**

All three reviewers suggested accepting this paper after discussing it with the authors during the rebuttal period. The AC agrees with the reviewers that the proposed semantic typography generation method is novel and interesting. The final recommendation is “Accept”.